# SocialFusion: Addressing Social Degradation in Pre-trained Vision-Language Models

**Hamza Tahboub**[1], **Weiyan Shi**[1], **Gang Hua**[2], **Huaizu Jiang**[1]

[1]*Northeastern University,* [2]*Amazon.com, Inc.*

{tahboub.h,we.shi,h.jiang}@northeastern.edu, ganghua@gmail.com

**Reviewed on OpenReview:** *https://openreview.net/forum?id=ofYhEoKIEx*

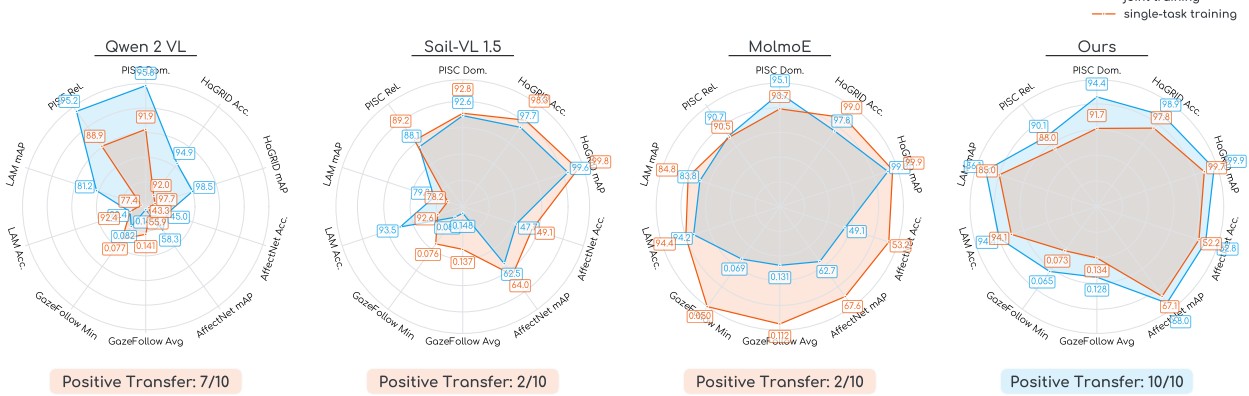

Figure 1: **SocialFusion achieves positive transfer across all visual social interaction understanding tasks.** Popular open-source VLMs suffer from social degradation, whereby their visual encoders have been degraded by the visual-linguistic pre-training, leading to negative transfer when jointly trained on multiple visual social interaction understanding tasks. SocialFusion addresses this limitation by using a frozen visual encoder and a minimal fusion architecture, achieving positive transfer on all tested task metrics while remaining competitive across the board.

## Abstract

Understanding social interactions from visual cues is a fundamental challenge for a socially competent AI. While powerful pre-trained vision-language models (VLMs) have shown remarkable general capabilities, they surprisingly struggle to unify and learn multiple social perception tasks simultaneously, often exhibiting negative transfer. We identify that this negative transfer stems from a critical issue we term "social degradation," whereby the general visual-linguistic pre-training process of VLMs impairs the visual encoder's ability to represent nuanced social information. We investigate this behavior further under two lenses: *decodability* through linear representation probing and *compatibility* through gradient conflict analysis, revealing that both play a role in the degradation, especially the former, which is significantly compromised in the VLM pre-training process. To address these issues, we propose SocialFusion, a unified framework that learns a minimal connection between a frozen visual encoder and a language model. Compared with existing VLMs, it exhibits positive transfer across all five social tasks, leveraging synergies between them to enhance overall performance and achieves comparable performance to task-specific state-of-the-art models on various benchmarks. Our findings suggest that current VLM pre-training strategies may be detrimental to acquiring general social competence and highlight the need for more socially-aware training paradigms.

## 1 Introduction

Social interactions with family members, friends, colleagues, and strangers are an important component of people's everyday lives. Understanding social interactions in visual media is a crucial capability for a visual

intelligence system (Lee et al., 2024b) and has broad applications in augmented reality (Geroimenko, 2023), human–robot interactions (Xu et al., 2021), and assistive technologies (*e.g.*, to help people with autism) (Iannone & Giansanti, 2023; Robila & Robila, 2020).

Social interaction understanding comprises various visual tasks, as shown in Fig. 2. Prior work has identified five important categories made up of purely visual tasks (Lee et al., 2024b): gesture analysis, which helps to understand conversation (Kendon, 1994; De Stefani & De Marco, 2019); gaze and attention analysis, which disambiguates people's intent and action (Fathi et al., 2012; Huang et al., 2015; Lukander et al., 2017; Emery, 2000); facial expression analysis, which provides direct insight into emotion, intent, and wishes (Horstmann, 2003); social situation analysis, inferring or reasoning about the social relationships, social norms, etc., in a scene; and conversation dynamics understanding, which involves tasks to understand speaker–addressee interactions, dialogue state tracking, etc. (Lee et al., 2024b).

Existing efforts toward solving such social interaction understanding tasks develop *specialist* models trained to solve one task at a time and feature task-specific architectures Lee et al. (2024b). While such models achieve good performance on many tasks, they are practically inconvenient for real-world use. A crucial step in building toward a general, socially competent system is to unify and jointly train diverse social interaction understanding tasks, especially if it is possible to achieve positive transfer by leveraging the synergies between different tasks.

General pre-trained vision-language models (VLMs) represent a natural candidate for unified social understanding, as they are known for a large number of diverse and general competencies (Li et al., 2024; Xu et al., 2025) due to their large-scale pre-training on similarly large and diverse datasets (Hurst et al., 2024; Deitke et al., 2024; Wang et al., 2024; Chen et al., 2025). We test several popular open-source VLMs of similar capacity: Qwen2-VL 2B (Wang et al., 2024), Sail-VL 1.5 2B (Dong et al., 2025), and MolmoE (Deitke et al., 2024), which is a mixture-of-experts model with 1B active parameters and 7B parameters in total. However, we find across the board that they are not able to foster consistent positive transfer between social tasks under parameter-efficient fine-tuning. In fact, as shown in Fig. 1, the models exhibit stark negative transfer, revealing an internal struggle to unify social competence.

We hypothesize that this failure stems from a phenomenon we term "social degradation," whereby VLM pre-training degrades visual encoders' representations for social tasks. To test this hypothesis, we measure the performance of visual encoders on various social tasks within different architectures, comparing them against their counterparts before visual-linguistic pre-training, which provides a fair and controlled environment for comparison. These experiments reveal that visual encoders perform worse on social interaction understanding tasks after VLM pre-training, demonstrating the social degradation. We investigate the mechanisms underlying social degradation under two lenses: linear *decodability* through linear representation probing and *compatibility* through gradient conflict analysis, revealing that both contribute to the degraded social representations, especially the former, which is significantly compromised in the VLM pre-training process.

To address these issues, we present SocialFusion, a *generalist* model to unify diverse visual social interaction understanding tasks using a minimal VLM. We learn the social fusion from scratch between a pre-trained visual foundation model and a language model to unify the interfaces of multiple visual social understanding tasks and perform them at a competitive level. We evaluate our approach on five social perception tasks in comparison with the three mentioned common pre-trained VLMs of similar parameter capacity to our model. To provide a fair comparison, we fine-tune all VLMs on the tasks with a LoRA adapter. We test the models on each task separately and with all tasks in conjunction. Unlike all other models, SocialFusion achieves positive transfer on all ten tested metrics (two per task), as visualized in Fig. 1.

Our contributions can be summarized as follows:

- Through controlled experiments comparing visual encoders before and after VLM pre-training, we identify "social degradation," whereby VLM pre-training degrades visual encoders' representations for social interaction understanding tasks.

- We investigate the mechanisms underlying social degradation through *decodability* via linear representation probing and *compatibility* via gradient conflict analysis, finding that reduced decodability is the primary factor.

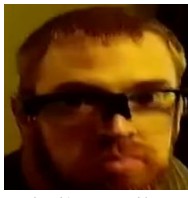 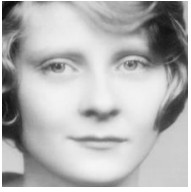 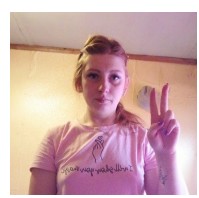 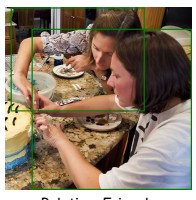 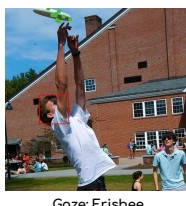

Looking at me: Yes     Expression: Contempt     Gesture: Peace     Relation: Friends / Domain: Intimate     Gaze: Frisbee (green points)

Figure 2: **Visual examples from each of the visual social interaction understanding tasks.** From left to right, the tasks are Ego4D Looking At Me (LAM), AffectNet facial expression recognition, HaGRIDv2 gesture recognition, PISC social situation analysis, and GazeFollow gaze target estimation.

- We propose SocialFusion, a generalist model that unifies diverse visual social interaction understanding tasks and allows us to explore the synergies of subtasks, advancing visual social interaction understanding as a whole.
- Our approach achieves positive transfer across all social tasks while setting new state-of-the-art results on the HaGRIDv2 and PISC benchmarks.

## 2 Visual Social Interaction Understanding Tasks

We consider five visual social interaction understanding tasks to establish broad coverage over the task taxonomy presented in (Lee et al., 2024b). Details of each task are presented below.

**Gesture analysis.** Understanding human gestures and changes in body language is essential for interpreting social interactions (Clough & Duff, 2020; Kendon, 1994; De Stefani & De Marco, 2019). Therefore, a successful social understanding model must have the ability to classify different gestures. For this task, we use the HaGRIDv2 dataset, which has 33 different classes of gestures in over one million images (Nuzhdin et al., 2024), to classify gestures such as the "peace" gesture depicted in Fig. 2.

**Gaze and attention.** Gaze position estimation can provide insights about an individual's intent and action (Fathi et al., 2012; Lukander et al., 2017) and enables better understanding of social interactions (Emery, 2000), as eye movement has been shown to reflect the human thought process (Yarbus, 2013). An example is shown in Fig. 2, where the man in the white shirt is focusing on the frisbee. The GazeFollow dataset (Recasens et al., 2015), containing 122,143 images with annotated gaze points, is a widely used benchmark for this task. Images with more than one person usually contain the gaze for each person. For that reason, the dataset also provides a bounding box annotation corresponding to each person. The GazeFollow test set has ten ground truth point annotations per sample from ten different annotators. The model's prediction is compared against these ten annotations to find the minimum and average distances.

**Facial expression recognition.** It has been established in psychology that there are several fundamental human facial expressions that are universal across cultures (Ekman & Friesen, 1971; Matsumoto, 1992). These expressions can reveal insights on emotional state, behavioral intentions, and wishes (Horstmann, 2003). An example with the "contempt" facial expression is shown in Fig. 2. In this paper, we use AffectNet (Mollahosseini et al., 2017) for model training and evaluation, which contains 450,000 images manually annotated with eight emotional categories.

**Conversation dynamics analysis.** This task involves understanding the flow of a conversation by modeling aspects like who is speaking to whom, the targets of participants' attention, and the evolving relationships of participants (Lee et al., 2024b). In this paper, we consider the "Looking at Me" (LAM) task from the Ego4D dataset (Grauman et al., 2022). It classifies three million samples of faces of participants of a conversation based on whether they are currently looking at the camera wearer, which gives insight into the conversation members' attention, demonstrated in Fig. 2 with a straightforward "Yes" classification.

**Social situation analysis.** This task includes understanding and inferring social relationships, answering social-reasoning questions, and other tasks that require an understanding of the social dynamic of a scene (Lee et al., 2024b). In the People in Social Context (PISC) benchmark (Li et al., 2017), relationships

are classified into predefined categories like "Friends," as illustrated in Fig. 2. PISC has two subtasks: classifying the relationship among specific six classes (family, couple, friends, commercial, professional, and no relationship) and classifying the domain of the relationship among three broad classes (intimate, non-intimate, and no relationship). The dataset contains 76,568 samples over 22,670 images, where an image may contain more than two people and thus multiple relationship pair samples. The input therefore also contains a pair of bounding boxes denoting the relationship of interest.

Together, these five tasks provide comprehensive coverage of visual social interaction understanding and enable unified modeling across diverse annotation types including classification labels, coordinate predictions, and bounding box inputs. The diversity of the tasks and their annotation formats presents both an opportunity for leveraging cross-task synergies and a challenge for unified modeling, which we address in the following section.

## 3 SocialFusion

### 3.1 Problem Formulation

The inputs to our SocialFusion model and baselines vary by task. All tasks' inputs include an image $x \in \mathbb{R}^{H \times W \times 3}$, where $H$ and $W$ are the image height and width, respectively. The gaze target estimation and social relationship classification tasks also include one and two bounding boxes per sample, respectively. Furthermore, in all tasks except gaze target estimation, the model predictions are output in plain-text tokens, as seen in the output of Fig. 3. The model differentiates between tasks based on the task prompts, which are formulated as `[task description]\n (a) [class 1] (b) [class 2] (c) ...`, and the model predicts the next token, which is a single token corresponding to the desired class. Specific task prompts are given in App. D. To handle gaze prediction, the model outputs a heatmap $h \in \mathbb{R}^{H_{out} \times W_{out}}$, indicating the 2D distribution of the gaze position, by applying a linear projection and upscaling the output image features.

### 3.2 The Social Degradation Issue in Pre-Trained VLMs

We first experiment with three widely used VLMs of similar capacity: the 2B parameter variation of Qwen2-VL (Wang et al., 2024), the 2B variation of Sail-VL 1.5 (Dong et al., 2025), and for additional architectural variety, MolmoE (Deitke et al., 2024), a mixture-of-experts model with 1B active/7B total parameters.

We fine-tune each of these VLMs using a low-rank adapter (LoRA, Hu et al., 2022) over the linear layers of the LLM module as a parameter-efficient and fair way to introduce new tasks. For each model, we consider two settings. We train each model both jointly, meaning training on all tasks simultaneously, and on each task separately. In joint training, we shuffle the datasets and randomly undersample them to the minimum size each epoch to avoid imbalanced training. Since we unified most tasks as plain-text generation, we are able to shuffle them within each batch. Otherwise, we randomly intersperse batches from the gaze heatmap prediction task with the regular text batches.

Table 1 shows the results of all three baseline VLMs on all tasks. None of the tested models are able to achieve consistent positive transfer in the joint training; rather, the models exhibit stark negative transfer in the majority of cases. In particular, Sail-VL 1.5 and MolmoE attain positive transfer on only two of ten metrics (one of five tasks) each. While Qwen2-VL achieves positive transfer on seven of the ten tested metrics, its performance is worse than the other baselines overall on most tasks, and three of ten metrics are still harmed. These results in conjunction show that these large pre-trained models are not able to easily generalize between these social tasks, instead driving them to compete when trained jointly.

We now investigate why the pre-trained VLMs are not able to unify these social tasks in harmony. We hypothesize that the problem lies in the models' visual encoders. In particular, we seek to measure of the impact of VLM pre-training on the encoders quantitatively. To this end, we compare each visual encoder before and after visual-linguistic pre-training. For Qwen2-VL, the original visual encoder is DFN ViT (Fang et al., 2023), for Sail-VL, it is SailViT (Dong et al., 2025), and for MolmoE, it is CLIP (Radford et al., 2021).

| Model | Joint Training | HaGRIDv2 | | PISC (mAP) | | LAM | | GazeFollow | | | AffectNet | |
|---|---|---|---|---|---|---|---|---|---|---|---|---|
| | | mAP ↑ | Acc ↑ | Domain ↑ | Relation ↑ | mAP ↑ | Acc ↑ | Min L2 ↓ | Avg L2 ↓ | AUC ↑ | mAP ↑ | Acc ↑ |
| Qwen2-VL (2B) | ✗ | 97.7 | 92.0 | 91.9 | 88.9 | 77.4 | **92.4** | **0.077** | **0.141** | – | 55.9 | 43.3 |
| | ✓ | **98.5** | **94.9** | **95.8** | **95.2** | **81.2** | 92.4 | 0.082 | 0.149 | – | **58.3** | **45.0** |
| MolmoE (1B/7B) | ✗ | **99.9** | **99.0** | 93.7 | 90.5 | **84.8** | **94.4** | **0.050** | **0.112** | – | **67.6** | **53.2** |
| | ✓ | 99.8 | 97.8 | **95.1** | **90.7** | 83.8 | 94.2 | 0.069 | 0.131 | – | 62.7 | 49.1 |
| Sail-VL 1.5 (2B) | ✗ | **99.8** | **98.3** | **92.8** | **89.2** | 78.2 | 92.6 | **0.076** | **0.137** | – | **64.0** | **49.1** |
| | ✓ | 99.6 | 97.7 | 92.6 | 88.1 | **79.3** | **93.5** | 0.086 | 0.148 | – | 62.5 | 47.7 |
| **SocialFusion (2B)** | ✗ | 99.7 | 97.8 | 91.7 | 88.0 | 85.0 | 94.1 | 0.073 | 0.134 | 93.4 | 67.1 | 52.2 |
| | ✓ | 99.9 | 98.9 | 94.4 | 90.1 | 86.1 | 94.4 | 0.065 | 0.128 | 94.0 | 68.0 | 52.8 |

Table 1: **Joint training results** on the validation sets of all considered tasks, except GazeFollow, which has no validation set and for which we use the test set as in prior work. All three tested pre-trained VLMs exhibit moderate to severe negative transfer when trained jointly on all tasks. Our SocialFusion performs at a competitive level with the VLMs, but it does not suffer from any negative transfer. Instead, it leverages the social synergies between them, resulting in an *improvement* in performance. In joint training, SocialFusion beats the three VLMs on all tasks except PISC.

In this initial experiment, we begin with a single social task: gaze target estimation. We test the VLM encoders, with versus without VLM pre-training, in Ryan et al. (2025)'s Gaze-LLE architecture, which produces state-of-the-art accuracy using DINOv2 on the GazeFollow dataset.

In Table 2, we provide a measure of the impact of VLM pre-training on the visual encoders of each of the three considered pre-trained VLMs. The effect here is significant, showing that all three VLMs suffer from a "social degradation" on a downstream visual social interaction understanding task. This initial evidence supports our social degradation hypothesis. We extend this analysis to all social tasks using our controlled experimental framework in Section 4.2.

| Visual Encoder | VLM Pre-training | Min L2 ↓ | Avg L2 ↓ | AUC ↑ |
|---|---|---|---|---|
| Qwen2-VL | ✗ | **0.076** | **0.140** | **93.8** |
| | ✓ | 0.114 | 0.184 | 90.9 |
| MolmoE | ✗ | **0.049** | **0.107** | **95.3** |
| | ✓ | 0.072 | 0.138 | 94.1 |
| Sail-VL 1.5 | ✗ | **0.049** | **0.108** | **95.3** |
| | ✓ | 0.051 | 0.110 | 95.2 |

Table 2: **VLM pre-training hurts visual encoders** for gaze target estimation under the Gaze-LLE (Ryan et al., 2025) architecture, under which we test the encoders of Qwen2-VL, MolmoE, and Sail-VL 1.5. All three encoders have degraded performance after VLM pre-training.

### 3.3 Investigating VLM Social Degradation

To better understand the cause of the "social degradation" issue, we analyze the visual encoders under two lenses: *decodability* and *compatibility*.

**Decodability.** We seek to understand the level of decodability of the embeddings produced by visual encoders before and after visual-linguistic pre-training. To that end, we use linear representation probes on the outputs of each encoder. Linear probing is a widely used method to investigate the inner workings of neural networks, allowing us to isolate the quality of each model's learned representations for a specific task while minimizing architectural biases (Saha et al., 2025; Belinkov, 2022; Alain & Bengio, 2017).

The goal here is not to improve performance, but to use linear decodability for controlled analysis. Our previous experiments suggest social degradation in the tested VLMs, but this result could be confounded by the language model and adaptation process. By using a simple linear probe on the raw visual features, we isolate the visual encoders from the language model entirely, allowing us to confirm whether the observed degradation is a true feature representation problem rather than an adaptation problem.

Specifically, we probe the representations on our $T = 5$ classification tasks (two distinct sets of classes under PISC). For each task $t \in \{1, \ldots, T\}$ and each visual decoder $f \in \{f_1, \ldots, f_n\}$, we train a linear classifier

$$W_t^* = \arg\min_{W_t} \mathcal{L}(W_t(\text{flatten}(f(X_t))) + b_t, Y_t),$$

where $W_t$ and $b_t$ are learned weights and biases that map the visual features to the number of classes for task $t$. $X_t$ and $Y_t$ are the input and ground-truth labels for task $t$, respectively. $\mathcal{L}$ is the cross-entropy loss.

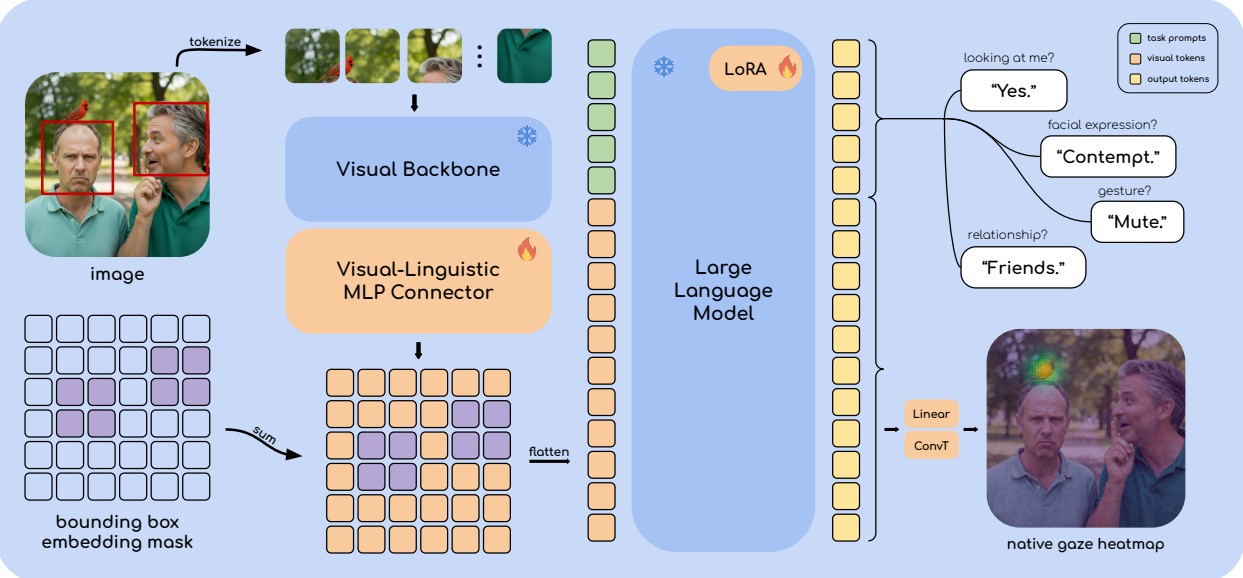

Figure 3: **Our SocialFusion architecture**. The input is an image and an optional set of bounding boxes. The bounding boxes are converted to a binary mask and multiplied by a learned embedding to create the bounding box embedding mask. The image is patchified and processed by the backbone. Then, the connector projects it to the embedding space of the LLM backbone, and the bounding box embeddings are added to this feature map. These updated feature maps are flattened and fed into the LLM alongside task-specific information. For classification tasks, the output logits are converted to tokens and interpreted as text. For gaze estimation, the final image features are linearly projected and upscaled to the desired 2D distribution.

**Compatibility.** In addition to the decodability of each task on its own, we also investigate whether visual-linguistic pre-training reduces the inter-compatibility of all tasks together. We measure this by calculating the gradient conflict between pre-trained and un-pre-trained pairs.

Formally, let $\mathcal{L}_t(\theta)$ be the loss function for training the entirety of task $t \in \{1, \ldots, T\}$ (as a single batch), parameterized by the weights $\theta$ of each encoder. Then, we can compute the gradient $g_t = \nabla_\theta \mathcal{L}_t(\theta) \in \mathbb{R}^d$ for all encoders on each task $t$. Following Chai et al. (2024), we use the angle $\phi_{ij}$ between gradients $g_i$ and $g_j$, comparing training on task $i$ against task $j$, to compute the *Gradient Conflict Degree (GCD)* between each pair of encoders, defined as

$$\text{GCD}(g_i, g_j) = 1 - \cos(\phi_{ij}) \in [0, 2],$$

where $\cos(\phi_{ij})$ is the cosine similarity of gradients $g_i$ and $g_j$. A higher *GCD* reflects more intense conflict between task gradients of tasks $i$ and $j$. Here, the weights we are comparing come from the active parameters of our model, which provides a controlled testbed for analyzing these behaviors in different visual encoders.

We will report the experimental results in Section 4.3 for both of these experiments, supporting our hypothesis that the VLM pre-training hurts the visual encoders. As a result, existing VLMs exhibit negative transfer when unifying different visual social interaction understanding tasks.

## 3.4 Proposed Model

In an effort to unify visual social interaction understanding tasks and achieve general social competence, we propose SocialFusion, a simple VLM framework that makes progress on this problem and allows us to analyze these behaviors in a controlled environment. SocialFusion principally consists of a pre-trained visual foundation model to encode the visual input, a language model to unify different task inputs and outputs with task prompts, and a connector to link these two components. This architectural choice is motivated by our observation that the standard VLM pre-training process itself degrades the visual encoder's ability to represent fine-grained social information. While other methods might focus on aligning potentially incompatible social tasks, we focus on this root cause. We propose that the most practical solution is to

avoid the degradation entirely by using a visual encoder that has not undergone the typical VLM pre-training pipeline and learning a minimal fusion with the language model.

Unlike other VLMs, SocialFusion supports 2D heatmap outputs indicating gaze positions by directly projecting and upscaling the final image features into a heatmap. Since some visual social tasks also include bounding boxes to refer to specific parts of the input, our model also natively accepts these as an additional input. Fig. 3 provides an overview of the model architecture, which we introduce in detail in the following.

**Visual encoder.** To extract a feature-rich representation from visual input, SocialFusion relies on a large pre-trained foundation model. We primarily use CLIP (Radford et al., 2021) as our visual encoder, which was trained to learn useful representations of images through natural language supervision. It has been successfully applied to diverse tasks in prior works (Lüddecke & Ecker, 2022; Ryan et al., 2025; Zhou et al., 2024; Zhang et al., 2022; Zhao et al., 2023). We freeze the weights of the visual encoder to reduce the number of trainable parameters. To allow for fine-grained understanding of the visual input, we use the $336 \times 336$ size input variant of CLIP and keep its weights frozen. With CLIP's $14 \times 14$ patch size, this results in a feature map $z \in \mathbb{R}^{24 \times 24 \times d_v}$ per image, where $d_v$ is the visual embedding dimension.

**Visual-linguistic connector.** The connector $C$ takes in the visual feature map $z$ and converts it to a sequence with the same embedding dimension $d_l$ as the LLM to align the visual and textual embeddings. We want the majority of the processing to take place in the LLM and learn a minimal connector between the two main components. Therefore, we opted for a straightforward three-layer MLP with a ReLU nonlinearity. We first flatten $z$ into a 576-length sequence. The first layer maps from $d_v$ to a hidden dimension $d_h$, the second layer maps again to $d_h$, and the third maps to $d_l$.

**Large language model (LLM).** The LLM of SocialFusion takes in a concatenated sequence $[\,p, C(z)\,]$, where $p$ is an embedded task-specific prompt that briefly describes the current task (see Appendix D for examples). This straightforward design promotes future work to similarly unify more multimodal inputs. For all tasks for which the ground truth can be represented in text, the model outputs the final model predictions as textual tokens, denoted by $\hat{y}$. In our experiments, we choose Llama 3.2 1B (Grattafiori et al., 2024) as the LLM. As we did in the case of the visual encoder, we freeze all of the weights of the LLM. However, to allow it to learn to process and analyze the visual inputs, we add a small LoRA adapter (Hu et al., 2022) to all of its linear layers, providing a fair comparison with the VLM baselines that we train in the same way.

**Bounding box embedding.** In the gaze target estimation and social relationship recognition tasks, the input also contains bounding boxes denoting people of interest. Inspired by Ryan et al. (2025), we incorporate bounding box information into the visual representations $z$ by adding a special learned position embedding, $p_{bbox}$, over features of patches that intersect with bounding boxes. Concretely, we construct a binary mask $M$ of the same dimension as the feature map $z$, where 1 indicates that a patch intersects with a bounding box and 0 means that it does not. Then, $z$ becomes

$$z = z_{initial} + M \odot p_{bbox}.$$

We apply this same procedure with the same learned embedding for all bounding boxes involved in different tasks. Future work on tasks with more than two bounding boxes may also consider using a different embedding vector for each box.

**Gaze heatmap projection.** SocialFusion must also output heatmaps for gaze target estimation. To maintain a largely unified architecture, we do not construct a heavy heatmap decoder. Instead, the LLM is responsible for learning the task. Then, we linearly project the model's output visual logits from the $24 \times 24 \times d_l$ feature map to $24 \times 24$. The final predicted heatmap $\hat{h}$ is then obtained by upscaling with a single transposed convolution and interpolating to the required heatmap size.

**Training Objective.** For the gaze heatmap output, the ground truth is a 2D Gaussian distribution (with $\sigma = 3$) around the actual gaze point, following prior work (Recasens et al., 2015; Chong et al., 2020; Ryan et al., 2025). We use a pixel-wise binary cross-entropy loss $\mathcal{L}_{heatmap}$ between the predicted and ground truth heatmaps. For all other tasks, since the ground truth is a sequence of textual tokens, we use a cross-entropy loss $\mathcal{L}_{llm}$ directly between the logits of the LLM and the ground truth tokens. The final loss is then

$$\mathcal{L} = \mathcal{L}_{llm}(\hat{y}, y) + \lambda \mathcal{L}_{heatmap}(\hat{h}, h).$$

| Visual | VLM | HaGRIDv2 | | PISC (mAP) | | LAM | | GazeFollow | | | AffectNet | |
| Encoder | Pre-training | mAP ↑ | Acc ↑ | Domain ↑ | Relation ↑ | mAP ↑ | Acc ↑ | Min L2 ↓ | Avg L2 ↓ | AUC ↑ | mAP ↑ | Acc ↑ |
| Qwen2-VL | ✗ | **98.9** | **95.5** | **92.9** | 85.6 | **85.6** | **93.1** | 0.087 | 0.153 | 92.7 | 61.3 | 45.1 |
| | ✓ | 98.4 | 94.5 | 91.4 | **87.3** | 79.0 | 92.6 | 0.123 | 0.190 | 88.4 | 60.4 | 44.5 |
| MolmoE | ✗ | **99.7** | **97.8** | **91.7** | **88.0** | **85.0** | **94.1** | 0.073 | 0.134 | **93.4** | **67.1** | **52.2** |
| | ✓ | 98.8 | 95.3 | 86.0 | 75.8 | 78.2 | 92.5 | 0.086 | 0.151 | 92.9 | 54.3 | 45.3 |
| Sail-VL 1.5 | ✗ | **99.6** | **97.7** | **89.6** | **86.6** | **79.8** | **93.2** | **0.059** | **0.118** | **94.1** | **60.4** | **45.0** |
| | ✓ | **99.6** | 97.5 | 88.8 | 85.8 | 78.7 | 92.4 | **0.059** | **0.118** | **94.1** | 59.6 | 44.5 |

Table 3: **Social degradation**. VLM pre-training hurts the visual encoders for social interaction understanding. We test the visual encoders of Qwen2-VL, MolmoE, and Sail-VL 1.5 in our SocialFusion architecture. For all models, pre-training leads to worse results for most considered social tasks.

Here, $\lambda$ is a hyperparameter to scale the weight of the heatmap loss depending on the number of tasks being trained jointly, as it would otherwise have the same impact as all other tasks combined.

Experimental results in Section 4.4 will show that our approach is able to avoid the negative transfer issue of other VLMs, achieving positive transfer on all tasks.

# 4 Experiments

## 4.1 Setup

**Implementation details.** We train all SocialFusion models with the AdamW optimizer (Loshchilov & Hutter, 2019) using an initial learning rate of $2e-4$, 500 warm-up steps, and cosine learning rate decay. We use a batch size of 32 for individual and joint training settings. Our LoRA adapters have a rank of 32; higher ranks did not improve performance in early experiments. For our visual-linguistic connector module, we use a hidden dimension of 4096. We experiment with three state-of-the-art VLMs: Qwen2-VL 2B (Wang et al., 2024), Sail-VL 1.5 2B (Dong et al., 2025), and MolmoE (Deitke et al., 2024), which is a mixture-of-experts model with 7 billion parameters in total and 1 billion active parameters. We use their recommended hyperparameters for training. Since these VLMs do not support native heatmap outputs, when making gaze predictions, we simply output the gaze points directly. This approach is viable for most VLMs because point data is included in their pre-training (Deitke et al., 2024; Wang et al., 2024), unlike our SocialFusion, which would not be able to work with points in text without additional training in that domain.

**Evaluation metrics.** For LAM, we report mAP and accuracy, as is done in the Ego4D paper (Grauman et al., 2022). The HaGRIDv2 paper only reports mAP for gesture detection (Nuzhdin et al., 2024), which we report, but since many models we test achieve near-perfect mAP scores, we also report accuracy to provide another point of comparison. Mollahosseini et al. (2017) report several metrics for AffectNet, including AUCPR and accuracy. In this setting, AUCPR is simply a slightly different way to calculate mAP and yields almost identical results, so we also report mAP and accuracy for this dataset, remaining comparable to those baselines. In the case of PISC, following Li et al. (2017), we report the mAP for each of the two classification subtasks: domain and relationship.

For GazeFollow, we report three metrics: 1) minimum L2, the minimum L2 distance between the prediction and all ten annotation points of the sample; 2) average L2, the average L2 distance between the prediction and the annotation points; and 3) AUC, the area under the ROC curve generated by using the heatmap prediction as a confidence score. We report only L2 distances for models that do not output heatmap distributions (*i.e.*, the pre-trained VLMs).

## 4.2 Further Evidence of Social Degradation in Pre-trained VLMs

To test our social degradation hypothesis comprehensively, we use SocialFusion as a controlled testbed, systematically comparing each VLM's visual encoder before and after VLM pre-training across all five social tasks. In particular, we swap the visual encoder inside SocialFusion with each of the VLMs' encoders,

allowing us to test each encoder in a fair setting within the SocialFusion framework. We keep the LLM constant and relearn the fusion from scratch for each encoder.

To achieve a quantitative measure of the impact of VLM pre-training as in Section 3.2, for each of the three pre-trained VLMs, we compare the use of their visual encoder before and after VLM pre-training. We report the results for these experiments in Table 3. For MolmoE, we observe a clear and pronounced degradation on all tested social tasks. Results on Qwen2-VL are more mixed in social relationship recognition, but on all other tasks, we can also see that the original encoder *before VLM pre-training* performs better. Similarly, Sail-VL's original encoder beats its pre-trained encoder on many tasks and matches performance on others, contrary to the expectation of pre-training leading to a stronger encoder.

These results in conjunction suggest that classical VLM pre-training on large amounts of general visual and textual data has a negative impact on the models' visual encoders for social interaction understanding.

### 4.3 VLM Social Degradation Investigation Results

Using the methods described in Section 3.3, we now analyze the social degradation property of the visual encoders in terms of *decodability* and *compatibility*.

**Decodability.** In Table 5, we present the validation results of our probes on each task–encoder pair. Qwen2-VL's un-pre-trained probe performance is better on every task except HaGRIDv2, Sail-VL 1.5 on every task except HaGRIDv2 and partially AffectNet, and MolmoE on every single task. These results strongly suggest that at least a significant part of the social degradation problem comes from reduced decodability of the visual features after general pre-training.

| Visual Encoder | VLM Pre-training | GCD |
|---|---|---|
| Qwen2-VL | ✗ | **0.941** |
| | ✓ | 0.961 |
| MolmoE | ✗ | **0.947** |
| | ✓ | 0.950 |
| Sail-VL 1.5 | ✗ | **0.931** |
| | ✓ | 0.934 |

Table 4: **Gradient conflict**. Joint training on visual social tasks using pre-trained VLM encoders leads to more conflict than the un-pre-trained versions.

**Compatibility.** We compute the cosine similarity between each task pair for each encoder at SocialFusion initialization and use these to calculate the *GCD* per encoder, which we report in Table 4. The results indicate that VLM pre-trained encoders slightly reduce inter-task alignment compared to the un-pre-trained versions. However, as noted by Chai et al. (2024), gradients are considered conflicting when $GCD > 1$. Since all tested encoders resulted in a $GCD < 1$, the task gradients remain aligned in this setting. Therefore, the pre-training slightly reduces alignment, but the differences are small and the gradients are not pushed to direct conflict.

### 4.4 Achieving Positive Transfer in Joint Social Interaction Understanding

As seen in Table 1, SocialFusion is the only model that achieves positive transfer on *all* tasks when trained jointly. In contrast to the baseline VLMs that exhibit negative transfer across most metrics (discussed in Section 3.2), SocialFusion leverages the synergies between social tasks to improve performance overall.

| Visual Encoder | VLM Pre-training | HaGRIDv2 | | PISC (mAP) | | LAM | | AffectNet | |
|---|---|---|---|---|---|---|---|---|---|
| | | mAP ↑ | Acc ↑ | Domain ↑ | Relation ↑ | mAP ↑ | Acc ↑ | mAP ↑ | Acc ↑ |
| Qwen2-VL | ✗ | 85.4 | 79.0 | **72.7** | **72.2** | **71.5** | **92.2** | **59.5** | **50.0** |
| | ✓ | **92.5** | **86.4** | 71.8 | 71.2 | 68.8 | 91.8 | 54.3 | 45.2 |
| MolmoE | ✗ | **96.0** | **90.9** | **70.4** | **68.9** | **76.3** | **92.5** | **60.7** | **50.1** |
| | ✓ | 85.3 | 79.7 | 62.5 | 57.8 | 48.6 | 90.1 | 41.4 | 40.2 |
| Sail-VL 1.5 | ✗ | 96.4 | 91.6 | **77.1** | **77.6** | **76.8** | **93.1** | 59.0 | **49.5** |
| | ✓ | **97.2** | **93.0** | 77.0 | 76.1 | 76.4 | 92.9 | **59.8** | 49.2 |

Table 5: **Linear probing of visual representations**. We use the features from each VLM's visual encoder before and after pre-training as the input to a probe to compare the linear decodability of each encoder. We see that the probes on encoders before VLM pre-training perform better overall, strengthening the social degradation finding and suggesting that it largely comes from reduced decodability of the features.

| Trained On | | HaGRIDv2 | | PISC (mAP) | | LAM | | GazeFollow | | | AffectNet | |
|---|---|---|---|---|---|---|---|---|---|---|---|---|
| Task One | Task Two | mAP ↑ | Acc ↑ | Domain ↑ | Relation ↑ | mAP ↑ | Acc ↑ | Min L2 ↓ | Avg L2 ↓ | AUC ↑ | mAP ↑ | Acc ↑ |
| AffectNet | LAM | – | – | – | – | 83.0 | 94.0 | – | – | – | 66.5 | 52.1 |
| AffectNet | HaGRIDv2 | 99.6 | 97.6 | – | – | – | – | – | – | – | 65.8 | 50.6 |
| AffectNet | GazeFollow | – | – | – | – | – | – | 0.067 | 0.129 | 93.5 | 64.3 | 50.8 |
| AffectNet | PISC | – | – | 93.2 | 87.7 | – | – | – | – | – | 65.7 | 50.9 |
| LAM | HaGRIDv2 | 99.5 | 96.8 | – | – | 82.1 | 94.1 | – | – | – | – | – |
| LAM | GazeFollow | – | – | – | – | 80.5 | 93.7 | 0.066 | **0.128** | 93.5 | – | – |
| LAM | PISC | – | – | 93.5 | 89.0 | 81.7 | 93.9 | – | – | – | – | – |
| HaGRIDv2 | GazeFollow | 99.5 | 97.1 | – | – | – | – | 0.070 | 0.130 | 93.7 | – | – |
| HaGRIDv2 | PISC | 99.6 | 97.6 | **94.4** | **90.8** | – | – | – | – | – | – | – |
| GazeFollow | PISC | – | – | 92.8 | 83.9 | – | – | 0.067 | 0.129 | 93.6 | – | – |
| Single-task training | | 99.7 | 97.8 | 91.7 | 88.0 | 85.0 | 94.1 | 0.073 | 0.134 | 93.4 | 67.1 | 52.2 |
| Joint training | | **99.9** | **98.9** | **94.4** | 90.1 | **86.1** | **94.4** | **0.065** | **0.128** | **94.0** | **68.0** | **52.8** |

Table 6: **Joint *vs.* pairwise synergies of different tasks**. We test the pairwise synergies of all tasks by jointly training our model from scratch on all such pairs. The columns represent the task being evaluated, and the rows represent different task pairings, including full joint training and individual training in the last two rows. We evaluate each pair on the tasks it was trained on, leaving other tasks as '–'. We can see some pairwise synergies in the table, improving tasks like PISC and GazeFollow. Some other tasks, like AffectNet and HaGRIDv2, are not improved, suggesting that a social training occurs when the number of tasks is increased, allowing the model to learn certain atomic social competencies shared between diverse tasks.

This stark difference in joint training performance supports our social degradation hypothesis. By using a frozen visual encoder that has not undergone VLM pre-training, SocialFusion avoids the representation degradation that hampers existing VLMs. The consistent positive transfer demonstrates that visual social interaction understanding tasks do contain exploitable synergies, but only when the visual representations are not compromised by the social degradation phenomenon.

In absolute terms, we also observe that our results beat or remain competitive with the baselines, which shows that even with the minimal training data of each task, SocialFusion is able to teach the LLM to use the pre-trained visual embeddings to solve complex tasks at the level of a generally pre-trained VLM. Importantly, when comparing only the joint models, SocialFusion beats all baselines on all datasets except PISC. These findings establish SocialFusion as a suitable architecture for learning different social interaction understanding tasks in conjunction by *leveraging their synergies* instead of *driving them to compete*.

The positive transfer results also validate our architectural choices. The minimal connector between frozen visual and language components preserves the quality of social representations while enabling cross-task learning, demonstrating that the key to unified social understanding lies not in larger or more complex models, but in avoiding the degradation that occurs during standard VLM pre-training.

## 4.5 Ablation on the Synergies of Different Tasks

To better understand the extent and nature of the positive transfer findings from Section 4.4, we conduct additional experiments to assess the synergy between all $\binom{5}{2} = 10$ possible pairings of tasks under our principal SocialFusion CLIP-based model.

As shown in Table 6, some pairs of tasks demonstrate some amount of synergy. For example, training on LAM and GazeFollow significantly improves the result on GazeFollow compared to single-task training, which is logical given the common element of deciphering gaze in both tasks, albeit in different ways. Similarly, training on HaGRIDv2 and PISC leads to improved results on PISC, potentially due to the large-scale nature of HaGRIDv2 allowing the model to learn more basic social competencies.

However, for some other tasks, such as AffectNet and HaGRIDv2, performance does not improve when trained with any one other task. For such tasks, where we do not observe a specific pairing that significantly improves performance even though we do observe complete positive transfer in the full joint training setting in Table 1, we conclude that jointly training SocialFusion on these diverse social tasks leads to a more general

| Model | HaGRIDv2 | PISC (mAP) | | LAM | | GazeFollow | | | AffectNet |
|---|---|---|---|---|---|---|---|---|---|
| | mAP ↑ | Domain ↑ | Relation ↑ | mAP ↑ | Acc ↑ | Min L2 ↓ | Avg L2 ↓ | AUC ↑ | Acc ↑ |
| SOTA | 89.4 | 87.0 | 79.5 | **81.0** | **93.0** | **0.041** | **0.099** | **95.8** | **66.32** |
| **SocialFusion (Ours)** | **99.8** | **94.2** | **88.5** | 76.0 | 92.0 | 0.065 | 0.128 | 94.0 | 52.8 |

Table 7: **Comparison with state-of-the-art methods** on different social interaction understanding tasks. SocialFusion sets a new SOTA on two tasks, is competitive on two others, and is behind on AffectNet.

social training effect that gives the model general *social competency*. This effect is only observed when the number of tasks is increased, forcing our model to learn deeper social insights.

## 4.6 Comparison to the State of the Art

Finally, to better situate our method's ability, we provide a comparison with the state of the art (SOTA) per task. We first focus on classification tasks. There have been many efforts to classify the social relationships in the two PISC subtasks (Yu et al., 2024; Li et al., 2022; 2020; Zhang et al., 2019; Goel et al., 2019; Wang et al., 2018). However, as seen in Table 7, SocialFusion significantly outperforms the SOTA approach (Berlincioni et al., 2025), highlighting the deep understanding that visual foundation models have of social scenes. On the test set, the SOTA is 87.0 mAP on PISC Domain and 79.5 on PISC Relations. The jointly-trained SocialFusion scores 94.2 mAP on PISC Domain and 88.5 mAP on PISC Relation, setting a new SOTA.

SocialFusion performs comparably or better than most baselines proposed in the original AffectNet paper (Mollahosseini et al., 2017) but is not at the level of more recent SOTA methods (Hassaballah et al., 2025; Zhou et al., 2025). HaGRIDv2 (Nuzhdin et al., 2024) is a recent extension over the HaGRID dataset (Kapitanov et al., 2024), so we could not find prior work on it besides the baselines presented in the work's paper. As shown in Table 7, the best baseline has an 89.4 mAP, which our jointly-trained model beats by a large margin with a mAP of 99.8 on the test set, setting the new state of the art. When it comes to LAM, there have also been several efforts dedicated to solving the task, with the recent state-of-the-art method using the InternVL image encoder and a Bi-LSTM network (Lertniphonphan et al., 2024) to achieve a strong result. Our model performs at a competitive level, a few points behind this recent method.

Finally, on GazeFollow (Recasens et al., 2015), we achieve competitive results to prior work, shown in Table 1, but do not beat the state of the art (Ryan et al., 2025) of 95.8 AUC, 0.041 minimum L2, and 0.099 average L2. Following Ryan et al. (2025), we also attempted to train SocialFusion using DINOv2 and were able to nearly match the state of the art, achieving a 95.3 AUC, 0.048 minimum L2, and 0.105 average L2, making it likely that this DINO-based model can match the SOTA with some hyperparameter tuning. However, this version of SocialFusion performed worse on all other tasks, which shows that the choice of visual encoder can have significant impact on certain tasks. For reference, we include the full results using DINOv2 as the visual backbone in Appendix A.

## 5 Related Work

**Visual social interaction understanding.** Building an intelligent system capable of understanding human social interaction is an underlying goal of many areas of research (Li et al., 2025; Recasens et al., 2015; Mollahosseini et al., 2017; Li et al., 2018; Lee et al., 2024a; Kapitanov et al., 2024; Ben et al., 2021). Nonetheless, efforts toward classifying the various tasks required for visual social interaction understanding are very recent (Lee et al., 2024b; Mathur et al., 2024).

Notably, Lee et al. (2024b) categorize visual social interaction tasks under two main categories: non-verbal social cue understanding and multimodal social cue understanding. These two categories are further split into six specific tasks. The former comprises gesture and body language analysis, gaze and attention analysis, and facial expression analysis, and the latter comprises multimodal emotion analysis, conversation dynamics analysis, and social situation analysis. This taxonomy lays the groundwork for building a robust, socially intelligent system, capable of analyzing visual and multimodal cues in a unified manner.

To our knowledge, there is no prior work that focuses on solving multiple visual social interaction under-standing tasks in conjunction. Our work fills this gap. We train our model on a wide coverage of these tasks, paving part of the way toward an intelligent system that can leverage its understanding of these lower-level social tasks in composition to understand and respond to more complex social situations.

**Vision-language models for multitask learning.** Traditionally, methods in multitask learning (MTL) (Zhang & Yang, 2021) train task-specific decoder heads or task-specific fine-tuned backbones to learn multiple tasks with some common parameters (Kokkinos, 2017; Xue et al., 2023). Since the development of strong general-purpose large language models and vision-language models (VLMs) (Zhang et al., 2024), a new paradigm for MTL has emerged, wherein this large model solves multiple tasks in a unified manner (Chen et al., 2023; Shen et al., 2024; Ding et al., 2022). VLMs can be repurposed on tasks unseen during training through in-context learning (ICL) (Dong et al., 2022) or fine-tuning (Zhang et al., 2024; Xing et al., 2024). ICL can be done in a few-shot fashion (Brown et al., 2020), where a few examples of how to solve the task are included in the prompt to the model, or a zero-shot fashion, where the model is given no examples and instead relies only on the task description. Zero-shot learning is particularly difficult in the multi-modal case (Cao et al., 2023) due to VLMs' task-agnostic training (Zhang et al., 2024), especially with smaller VLMs on difficult tasks, and few-shot learning is not practically scalable with multiple datasets and long sequences of visual tokens in each example.

Therefore, adapting the powerful pre-training of VLMs on downstream tasks through transfer learning has become common for difficult tasks (Zhang et al., 2024; Xing et al., 2024). Full or even partial direct fine-tuning, however, is often also impractical with these large models, giving rise to efficient methods like adapters (He et al., 2021; Xing et al., 2024), prefix-tuning (Li & Liang, 2021), prompt tuning (Lester et al., 2021; Jia et al., 2022), low-rank adaptation (LoRA) (Hu et al., 2022), and other variants (Xing et al., 2024).

The choice of a parameter-efficient fine-tuning method does not, by itself, resolve all underlying challenges of MTL. A key issue is negative transfer, where training on multiple tasks simultaneously degrades performance compared to training on tasks individually. This can stem from gradient conflict, where parameter updates for different tasks pull in opposing directions (Zhang & Yang, 2021). Advanced MTL methods have been proposed to address this, such as PCGrad (Yu et al., 2020), which projects task gradients to remove any conflicting components, or other approaches that dynamically re-weight task losses or align gradients (Chen et al., 2018; Liu et al., 2021). However, these methods primarily address optimization-level interference. While they could help slightly alleviate the problem due to the small increase in gradient conflict we observed in VLM visual encoders, they are not designed to solve the core issue identified in our work: a fundamental representation degradation. Thus, merely aligning gradients cannot fully address the social degradation in the pre-trained visual encoders.

To provide a fair comparison, we compare our SocialFusion method with VLMs fine-tuned using a LoRA adapter, which is the same technique we use to fine-tune the underlying LLM behind SocialFusion. In particular, we experimented with three widely used VLMs of similar capacity: the 2B parameter variation of Qwen2-VL (Wang et al., 2024), the 2B variation of Sail-VL 1.5 (Dong et al., 2025), MolmoE (Deitke et al., 2024), a mixture-of-experts model with 1B active parameters and 7B parameters in total.

## 6 Conclusion

In this work, we identified and investigated "social degradation," a phenomenon whereby VLM pre-training degrades visual encoders' representations for social interaction understanding tasks. Through controlled experiments across three popular VLMs, we demonstrated that visual encoders consistently perform worse on social tasks after VLM pre-training. Our analysis revealed that this degradation primarily stems from reduced linear decodability of social features, with gradient conflict playing a secondary role.

To address social degradation, we proposed SocialFusion, a unified framework that learns minimal connec-tions between frozen visual encoders and large language models. It achieves *positive transfer across all five social tasks*, demonstrating that cross-task synergies can be effectively leveraged when social degrada-tion is avoided. Our approach sets new state-of-the-art results on HaGRIDv2 and PISC benchmarks while remaining competitive across all tasks.

These findings have important implications for the VLM community. Current VLM pre-training strategies, while effective for general capabilities, may inadvertently harm specialized social interaction understanding. The social degradation phenomenon suggests that the visual representations learned during large-scale multimodal pre-training may not preserve the visual cues necessary for social perception tasks.

**Limitations and future directions.** Our analysis focuses on visual encoders across five social tasks, providing a foundation for investigating social degradation in other VLM components and specialized domains. While SocialFusion's minimal architecture with frozen components effectively demonstrates positive transfer, exploring more sophisticated architectures could further improve performance, particularly on tasks like AffectNet where current results lag behind recent methods. Furthermore, our model is not designed nor intended for general, unseen tasks, so further research should investigate mitigating the social degradation in generalist models directly. Our investigation identifies reduced decodability as the primary factor in social degradation, opening important questions about the specific mechanisms during VLM pre-training that cause this effect.

These findings point to several promising research directions. First, developing VLM pre-training strategies that preserve social representations could eliminate the need for specialized architectures altogether. The social degradation phenomenon suggests investigating whether similar effects occur in other specialized domains, potentially revealing fundamental insights about large-scale multimodal pre-training trade-offs. Second, SocialFusion's consistent positive transfer across social tasks demonstrates the feasibility of unified social understanding, making it an ideal foundation for tackling more complex compositional social reasoning tasks such as social strategy games and advanced human-computer interaction scenarios. Finally, incorporating socially-aware curricula into VLM pre-training could address the root causes of social degradation while maintaining general capabilities.

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

| Encoder | HaGRIDv2 | | PISC (mAP) | | LAM | | GazeFollow | | | AffectNet | |
|---|---|---|---|---|---|---|---|---|---|---|---|
| | mAP ↑ | Acc ↑ | Domain ↑ | Relation ↑ | mAP ↑ | Acc ↑ | Min L2 ↓ | Avg L2 ↓ | AUC ↑ | mAP ↑ | Acc ↑ |
| CLIP | **99.9** | **98.9** | **94.4** | **90.1** | 86.1 | 94.4 | 0.065 | 0.128 | 94.0 | **68.0** | **52.8** |
| DINOv2 | 99.4 | 96.7 | 90.7 | 64.1 | **87.3** | **94.9** | **0.048** | **0.105** | **95.3** | 59.1 | 47.9 |

Table 8: Comparing our standard CLIP encoder with DINOv2 in the SocialFusion architecture. DINOv2 achieves better results on two gaze-related tasks but is significantly weaker on other tasks.

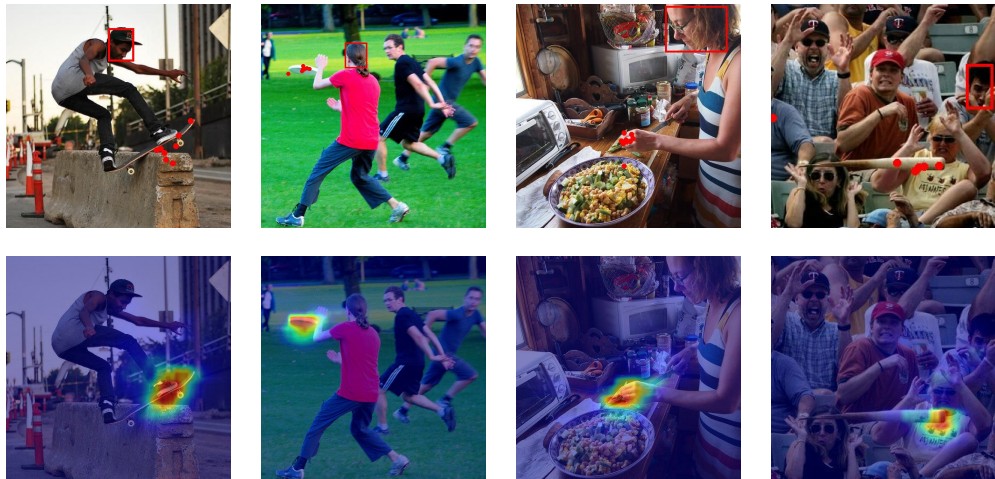

Figure 4: Examples of SocialFusion's outputs on the gaze estimation task (GazeFollow). The top row includes examples of input images from the dataset as well as the ground truth annotations (one point per annotator). The bottom row is our model's output heatmap distribution per image.

## A    DINOv2 as a Visual Backbone

For completeness, we report in Table 8 results when training SocialFusion jointly with DINOv2 as the visual encoder instead of CLIP. The results when training individually on GazeFollow are a few points better than CLIP, and results on LAM are also slightly better. However, results on other tasks are marginally to significantly worse, which establishes CLIP as a better choice overall.

## B    Negative transfer in InternVL3

We present in Table 9 results from the experiments associated with Table 1 for the VLM InternVL3 2B model. Similar to other tested VLMs, this newer model suffers from negative transfer on most tasks, exhibiting positive transfer on only one metric.

## C    Visual Results

In Figures 4 and 5, we showcase outputs from our main SocialFusion model with the CLIP backbone.

| Model | Joint Training | HaGRIDv2 | | PISC (mAP) | | LAM | | GazeFollow | | | AffectNet | |
|---|---|---|---|---|---|---|---|---|---|---|---|---|
| | | mAP ↑ | Acc ↑ | Domain ↑ | Relation ↑ | mAP ↑ | Acc ↑ | Min L2 ↓ | Avg L2 ↓ | AUC ↑ | mAP ↑ | Acc ↑ |
| InternVL3 (2B) | ✗ | **99.8** | **98.5** | **95.5** | 91.9 | **82.3** | **93.8** | **0.070** | **0.132** | − | 65.9 | 51.8 |
| | ✓ | 99.5 | 97.3 | **95.5** | **92.1** | 81.1 | 93.7 | 0.081 | 0.145 | − | 63.2 | 47.9 |
| **SocialFusion (2B)** | ✗ | 99.7 | 97.8 | 91.7 | 88.0 | 85.0 | 94.1 | 0.073 | 0.134 | 93.4 | 67.1 | 52.2 |
| | ✓ | **99.9** | **98.9** | **94.4** | **90.1** | **86.1** | **94.4** | **0.065** | **0.128** | **94.0** | **68.0** | **52.8** |

Table 9: Like other VLMs, InternVL3 does not exhibit positive transfer in the joint setting.

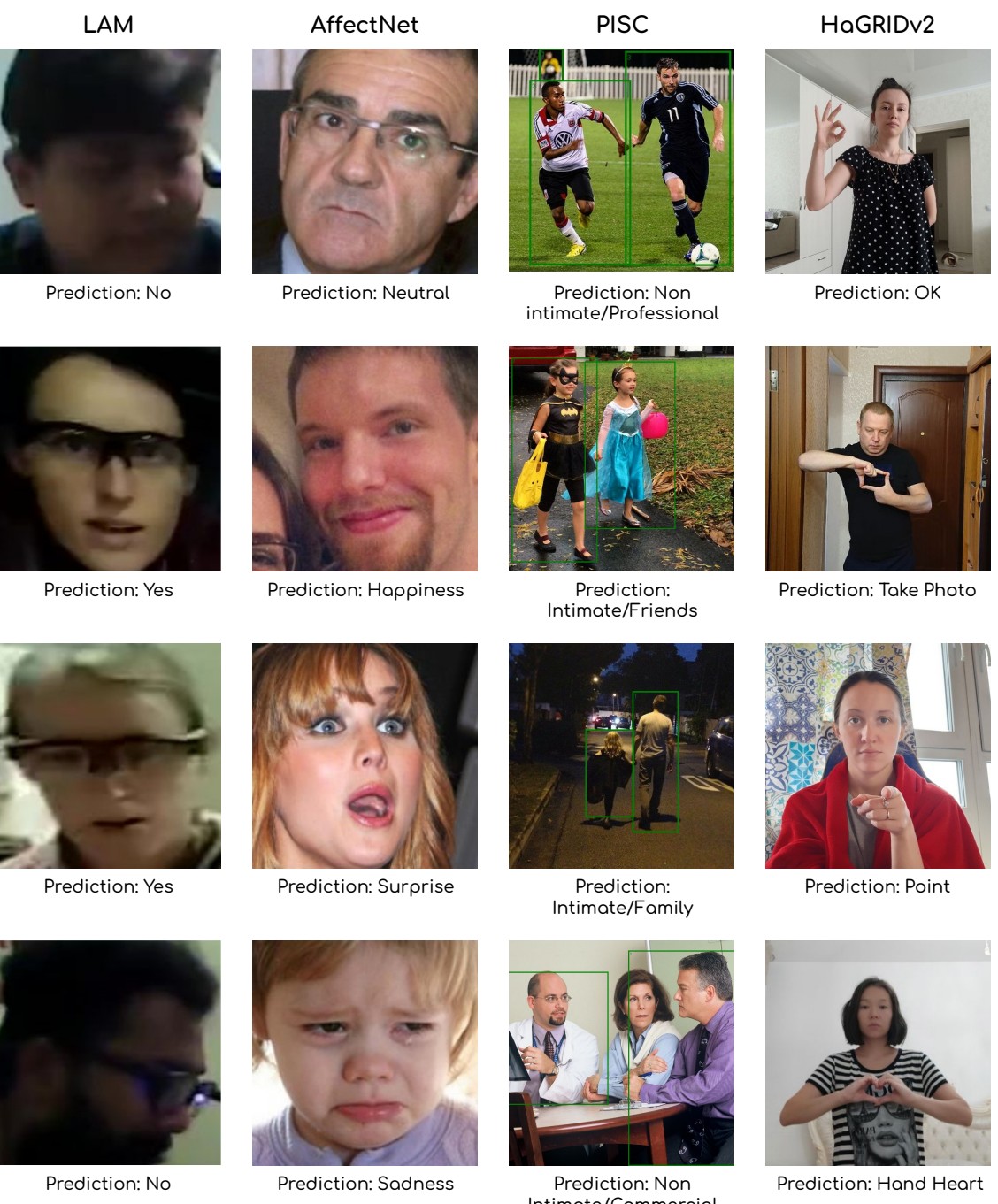

Figure 5: Examples of SocialFusion's successful outputs on each of the classification tasks.

## D  Task-Specific Prompts

We provide the model with the prompts in figures 6 through 10 in order to give it an initial idea of the task definitions before fine-tuning. Each task contains an image as input, as in the examples with each prompt.

You will be given an image containing a person. You must identify the facial expression of this person. The following is a list of all possible expressions that you can choose from:
(a) neutral (b) happiness (c) sadness (d) surprise (e) fear (f) disgust (g) anger (h) contempt
Identify the expression of the person in the image and respond directly with only the expression's lowercase label (e.g. 'a', 'c', 'h') with no additional text or reasoning.

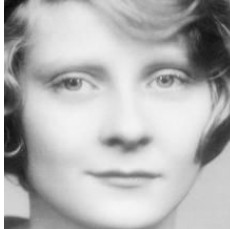

Expression: Contempt

Figure 6: Task prompt and example input/output for AffectNet.

You will be given an image containing at least one person. You will also be given a bounding box specifying a certain person's head. Your task is to predict where this person is looking. You are expected to identify the exact point they are looking at. Please provide your answer as a coordinate '(X,Y)', where both X and Y are in the range [0, 1000). For example, if the person appears to be looking at something exactly in the center of the image, the answer would be '(500,500)'. You must be as exact as possible. To repeat, please output ONLY '(X,Y)' with no other reasoning or explanation whatsoever.

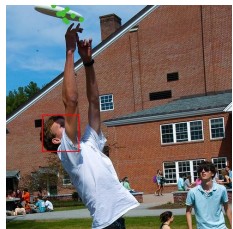

Gaze: Frisbee
(green points)

Figure 7: Task prompt and example input/output for GazeFollow.

You will be given an image containing several people and two bounding boxes around two of those people. Your task is to classify the nature of the social relationship between these two people. Your choices are
(a) intimate (b) not intimate (c) no relation
Your response should simply be your best estimate of the two individuals' social relation as a single letter (one of 'a', 'b', or 'c') with no additional text or reasoning.

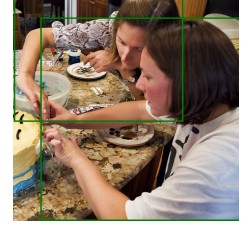

Relation: Friends
Domain: Intimate

You will be given an image containing several people and two bounding boxes around two of those people. Your task is to classify the specific social relationship between these two people. Your choices are
(a) friends (b) family (c) couple (d) professional (e) commercial (f) no relation
Your response should simply be your best estimate of the two individuals' social relation as a single letter (e.g. 'a' or 'f') with no additional text or reasoning.

Figure 8: Task prompt and example input/output for PISC Domain and Relationship.

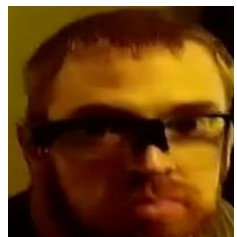

Looking at me: Yes

You will receive a single image with a cropped face. You must respond 'Yes' if the face in the image is looking at the camera wearer and 'No' otherwise. There will always be a face in the image, but since it is cropped from a larger image, it may be low resolution and unclear. Simply respond according to your best estimate. The only valid responses are 'Yes' and 'No', with no additional explanation.

Figure 9: Task prompt and example input/output for LAM.

You will be given an image containing a person. You must identify the gesture this person is performing. The following is a list of all possible gestures that you can choose from:
(a) phone gesture with only the thumb and pinkie out (b) thumbs up gesture (c) thumbs down gesture (d) a closed fist gesture (e) gesture with only the index finger out (f) gesture with only the index, middle, and ring fingers out (g) gesture with only the thumb, index, and middle fingers out (h) gesture with only the index, middle, and pinkie fingers out (i) gesture with all fingers except thumb out (j) gesture with index finger over lips (k) gesture with thumb and index touching and all other fingers out (l) gesture with hand splayed out, exposing the palm (m) gesture with only the index and middle fingers out, palm facing camera (n) gesture with only the index and middle fingers out, back of hand facing camera (o) gesture with only index and pinkie fingers out (p) gesture with all fingers out and touching each other closely, palm facing camera (q) gesture with all fingers out and touching each other closely, back of hand facing camera (r) gesture with index and middle fingers out and touching each other closely, palm facing camera (s) gesture with index and middle fingers out and touching each other closely, back of hand facing camera (t) gesture with all fingers out in a claw form, as if grabbing something (u) gesture with fingers arranged in a circle, as if gripping a pole (v) gesture pointing with the index finger at the camera (w) gesture with only the pinkie out (x) gesture with only the middle finger out (y) gesture with the index and middle fingers out and touching, and the thumb out perpendicular to them (z) gesture with only the thumb and index fingers out perpendicular to each other (aa) thumb index gesture with both hands at the same time (ab) gesture with both hands pressed together, as if praying (ac) gesture with both hands flat touching each other perpendicularly with one touching the other's palm (ad) thumb index gesture with both hands touching each other in the form of a rectangle (ae) both hands closed and arms crossed against each other (af) both hands coming together in the shape of a heart with all fingers forming the shape (ag) only thumb and index fingers of both hands forming the shape of a heart, with other fingers pointing up

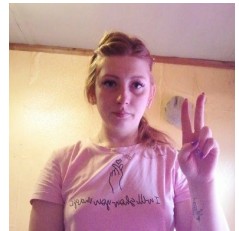

Gesture: Peace

Identify the gesture that the person is performing and respond directly with only the gesture's lowercase label (e.g. 'a', 'j', 'ag') with no additional text or reasoning.

Figure 10: Task prompt and example input/output for HaGRIDv2.

