# OpenReview forum: "SocialFusion: Addressing Social Degradation in Pre-trained Vision-Language Models"
_TMLR — Accepted by TMLR_

### Review · Reviewer_bW6a · 2025-11-06

**Summary Of Contributions:**

The paper's primary contribution is the analysis of "social degradation," the finding that standard VLM pre-training can impair a visual encoder's ability to represent social information. The authors support this claim through experiments showing degraded performance on social tasks after VLM pre-training, which they link to reduced feature decodability. As a solution, they propose SocialFusion, an architecture using frozen visual and language backbones with a small trainable connector between the two.

A strength is that SocialFusion is the first to demonstrate consistent positive transfer across all five social tasks the authors tested, in contrast to the negative transfer seen in standard VLM fine-tuning. However, a significant weakness is the paper's limited baselines: while it shows its method is superior to a naive fine-tuning approach, it fails to compare against any advanced multitask learning or parameter-efficient fine-tuning methods that are also designed to mitigate the observed gradient conflicts and negative transfer, leaving it unclear if the proposed architecture is a necessary solution or just one of many possible ones.

**Audience:**

Yes

**Audience Explanation:**

Improving multimodality and evaluating the weaknesses of multimodal language models are both highly relevant areas of research.

**Claims And Evidence:**

Yes

**Claims Explanation:**

I ultimately chose yes because the authors' primary contribution (in my opinion) is the identification and quantification of social degradation, and they provide clear empirical evidence for this phenomenon (Tables 2 & 3). Their analysis further supports this claim by investigating the root causes, such as reduced feature decodability (Table 5). The proposed "SocialFusion" architecture is then presented as a solution, and Table 1 shows it does achieve positive transfer.

However, I'll also reiterate that the above evidence does not support a potential implicit claim that SocialFusion is the best or only solution. The authors do not baseline their method against alternatives from multi-task learning or adapter-based PEFT literature that are also designed to mitigate negative transfer.

**Requested Changes:**

- Critical for acceptance: **Deepen the related work section** and discuss the other possible ways to solve this problem. For example, the authors mention gradient conflicts as a reason for degradation, and a classic way to solve this is PCGrad or variants of it, some of which have been cited by the authors.
- Critical for acceptance: Give readers more intuition as to why your method is the most reasonable or practical choice. (I think it is indeed reasonable, but please make this explicit.) Discuss the weaknesses of PEFT and gradient alignment based methods, leading to the authors' more architecture-based approach.
- Strengthen work: Consider adding baselines like PCGrad or MoCoGrad or PEFT methods.

---

> ### Author Response · Authors · 2025-11-21
> **Response**
>
> We thank the reviewer for the thoughtful review and for recognizing the value of our social degradation finding.
>
> ---
>
> > while [the paper] shows its method is superior to a naive fine-tuning approach, it fails to compare against any advanced multitask learning or parameter-efficient fine-tuning methods that are also designed to mitigate the observed gradient conflicts and negative transfer
>
> Thank you for raising this point. We would like to clarify that to provide a fair comparison while remaining computationally feasible, all our fine-tuning experiments were run with a LoRA adapter and not full fine-tuning. We have now made this clearer in the revised paper. However, your point still stands that we do not compare with more techniques to further validate the findings. We believe that the consistent result under the controlled setting we provided over three different VLMs of similar capacity justifies the claims we have made regarding the degradation, so we have followed your suggestion of expanding the related work section to discuss some of these approaches in more detail, and we will leave further experimentation and validation under this direction to future work.
>
> > the above evidence does not support a potential implicit claim that SocialFusion is the best or only solution
>
> We hope now with the addition from your above concern that helped us situate SocialFusion as the most practical approach, as well as our existing discussion on potential future work in the conclusion, that the implication that our method is the only solution has been sufficiently tempered.
>
> > Give readers more intuition as to why your method is the most reasonable or practical choice. (I think it is indeed reasonable, but please make this explicit.)
>
> We now also include further intuition in the discussion as to why our approach is reasonable and practical. Briefly, we explain that the issue is not that the social tasks are inherently incompatible (which gradient alignment would fix), but that the VLM pre-training process itself has already degraded the visual encoder's ability to represent social information. Methods in gradient alignment may be beneficial, but they would not address this root cause. Instead, the most practical solution is to avoid the degraded component entirely, using an encoder that has not undergone the VLM pre-training pipeline.
>
> ---
>
> We hope we were able to address your core concerns. Please let us know if you would like any additional clarifications.

---

### Review · Reviewer_v9eN · 2025-11-07

**Summary Of Contributions:**

This paper demonstrates the difficulty of realizing transfer learning benefits in social understanding tasks, isolates part of the problem to the visual encoder, and addresses the problem by freezing the visual encoder.

This paper is focused on a set of five different but overlapping visual social interaction understanding tasks: gesture classification, gaze estimation, facial expression recognition, "Looking at Me" binary classification, and interpersonal relationship classification. Each task has 2 metrics associated with it, so there are 10 metrics total. Since the images and tasks are similar there is hope that transfer learning would improve performance.

The first set of experiments demonstrates the "social degredation" transfer learning problem. Each task is expressed as a prompt and with the expected output being one of a fixed set of tokens per classification problem. Gaze position is estimated by attaching a head that is trained to output heatmaps of gaze location. These models are (A) finetuned on each of the 5 tasks individually and (B) finetuned on the 5 tasks jointly. Positive transfer occurs when performance in case B is higher than in case A. Sail-VL and MolmoE demonstrate positive transfer on 2 of 10 metrics while Qwen 2 VL demonstrates positive transfer on 7 of 10 metrics.

A VLM architecture called SocialFusion is proposed. It uses a small visual linguistic adapter layer to adapt a visual backbone to an existing LLM's (Llama 3.2) embedding space. This adapter also incorporates bounding box input as an embedding added to each visual token where the box is present. It outputs text for classification tasks and heatmaps for the gaze task. Only the adapter layer and a small LoRa adapter on the LLM are finetuned, NOT the visual encoder itself.

The second set of experiments tests the effect of VLM pre-training on visual encoder in isolation. It trains 3 variants of the SocialFusion model using the 3 different visual encoder backbones from Qwen 2 VL, Sail-VL, and MolmoE. By comparing performance of the off the shelf visual encoder (without VLM pre-training) to the visual encoder after VLM pre-training the experiments show clearly that VLM pre-training hurts visual encoder performance.

The SocialFusion architecture shows positive transfer on 10 of 10 metrics.

Additional experiments:
* "Decodability" is the ability of a visual encoder to perform well with just a linear probe. Linear probes applied to the visual encoders for each of the 10 metrics show a similar transfer learning problem when comparing VLM pre-trained visual encoders to the original visual encoders themselves.
* Gradient are slightly less similar across different tasks for VLM pre-trained visual encoders, but the significance of this is unclear.
* Social Fusion provides a new SOTA result for gesture analysis (HaGRIDv2) and social situation analysis (PISC).
* A task-pairwise transfer learning analysis demonstrates a similar issue as learning all 5 tasks at once, except that transfer is observed between two pairs of tasks that are intuitively more similar.

The paper claims that
* It demonstrates "social degredation" in VLMs
* "Decodability" is the primary issue in "social degredation" transfer learning
* The SocialFusion architecture advances social interaction understanding
* SocialFusion achieves positive transfer and sets new SOTA on HaGRIDv2 and PISC

**Audience:**

Yes

**Audience Explanation:**

Researchers and engineers interested in social interaction understanding would be interested to know transfer learning can be demonstrated clearly and addressed to an extent through this approach in their domain.

Researchers who study transfer learning will find this interesting as a clean study that demonstrates the problem in a new domain.

Researchers and engineers simply interested in the state of the art in VLMs will not be very interested. The newest model used is from December 2024, which is relatively old. More recent models have demonstrated superior performance (e.g. Qwen 2.5 VL in Feb '25) and it is not clear how well these results would transfer to those models, especially given that Qwen 2 exhibits the problem significantly less (Figure 1).

**Claims And Evidence:**

Yes

**Claims Explanation:**

I think this is a valuable contribution and would like to accept it, but I have an issue with claim 2 as described below that I think can be address as a matter of presentation.

1. "Through controlled experiments comparing visual encoders before and after VLM pre-training, we identify “social degradation,” whereby VLM pre-training degrades visual encoders’ representations for social interaction understanding tasks."
    * This is clearly demonstrated through the notion of positive transfer. (Table 1)
2. "We investigate the mechanisms underlying social degradation through decodability via linear representation probing and compatibility via gradient conflict analysis, finding that reduced decodability is the primary factor."
    * This is not cleary demonstrated. The paper does this investigation, but I'm not even sure what the claim "reduced decodability is the primary factor" means much less whether it is true. I think the paper includes a nice investiation of the impact of the visual encoder, including Tables 3 and 5. This clearly shows that VLM pre-training of the visual encoder hurts performance, but that is more directly shown without linear probing (i.e., by Table 3). Furthermore, it could still be the case that joint finetuning of the LLM itself (via LoRa) also demonstrates some performance degredation, and it is not clear how this would compare to the impact of the visual encoder.
3. "We propose SocialFusion, a generalist model that unifies diverse visual social interaction understanding tasks and allows us to explore the synergies of subtasks, advancing visual social interaction understanding as a whole."
    * This is clearly demonstrated. The SocialFusion supports effective experiments.
4. SocialFusion achieves positive transfer and sets new SOTA on HaGRIDv2 and PISC
    * This is clearly demonstrated in Table 1 and Table 7.

**Requested Changes:**

Throughout: the tables and figures aren't well aligned with where they are discussed in the text.

Section 4.3: This section concludes that the GCD is small and insignificant. I would like to see that elaborated some. From one perspective there is no baseline for me to calibrate GCD magnitude against. Even though the scale is from 0 to 2 it could still be that a change of .003 is significant. Maybe the paper could reference a result from Chai et al. (2024) to calibrate these numbers. From another perspective a change like that could be statistically significant. Measuring standard deviation across gradients from different splits of the whole dataset could help calibrate this. If 0.003 is within standard deviation then maybe not so significant, but if its outside then maybe it is.

---

> ### Author Response · Authors · 2025-11-21
> **Response**
>
> Thank you for the insightful and constructive feedback. We are encouraged that the reviewer finds the majority of our claims to be well substantiated, and we appreciate the opportunity to clarify our work based on the points raised.
>
> ---
>
> > I'm not even sure what the claim "reduced decodability is the primary factor" means much less whether it is true. I think the paper includes a nice investigation of the impact of the visual encoder, including Tables 3 and 5. This clearly shows that VLM pre-training of the visual encoder hurts performance, but that is more directly shown without linear probing (i.e., by Table 3).
>
> This is a good point, and we have worked to improve our discussion on our motivation for these experiments in the revised manuscript. We now explain further that our motivation behind testing what we call decodability on top of the investigation of the impact of the visual encoders within the VLMs is to confirm that the social degradation observed in those initial experiments still holds up without the complex LLM. By measuring how easily a simple model can decode the social information from the raw visual features, we reduce the potential confounding variables and confirm more strongly that the visual encoders themselves are compromised; the degradation is a feature representation problem, not just an adaptation problem.
>
> > Furthermore, it could still be the case that joint finetuning of the LLM itself (via LoRa) also demonstrates some performance degredation, and it is not clear how this would compare to the impact of the visual encoder.
>
> As mentioned above, this is part of the reason behind our decodability experiments. Even though we specifically controlled for the potential influence of the LLM by keeping it constant in all the experiments in Table 3, there remains a chance that the degradation can be caused by a failure of the LLM to adapt via LoRA. Hence, we simplify the setting considerably in Table 5 to completely isolate the visual encoders.
>
> > Throughout: the tables and figures aren't well aligned with where they are discussed in the text.
>
> We have improved this issue in the updated manuscript, and we will make sure that the figures are as well aligned as possible in the final revision.
>
> > Section 4.3: This section concludes that the GCD is small and insignificant. I would like to see that elaborated some. From one perspective there is no baseline for me to calibrate GCD magnitude against. Even though the scale is from 0 to 2 it could still be that a change of .003 is significant.
>
> This is also an important point. In the context of gradient conflict and alignment, and as explained by Chai et al., the gradients of two tasks are considered to be conflicting when $GCD > 1$. Therefore, we claim that this issue is relatively insignificant because all three settings we tested did not exhibit direct gradient conflict (the gradients are still on the same side of the plane in all cases). While the VLM training did slightly reduce this alignment, they do ultimately remain aligned, and the delta is rather small, so we can say that in comparison to the much starker difference observed in the decodability experiments, compatibility should not be a major focus. We have improved the discussion in the paper to make this reasoning more explicit.
>
> ---
>
> We thank the reviewer again and hope that we were able to address your concerns to your satisfaction.

---

> ### Comment · Reviewer_v9eN · 2025-12-12
> **Author clarifications addressed my concerns**
>
> My concern that claim 2 was not clearly demonstrated is addressed. After looking at this again and reading the author responses my thinking was not very clear, and the the first two author responses above (which will also be in the final paper) helped me understand that.
>
> The other concerns were minor but addressed by the author responses.
>
> I updated my answer to the first question about claims to reflect this.

---

> > ### Author Response · Authors · 2025-12-13
> > **Response**
> >
> > We are grateful to hear that you agree with the reasoning behind our claims. Thank you again for your review, which we believe has improved the clarity and quality of the paper.

---

### Review · Reviewer_1YrF · 2025-11-11

**Summary Of Contributions:**

In this paper, the authors observed that performance degrades of current VLMs for social interaction understanding tasks by applying controlled experiments. The authors assess the decodability and the task compatibility of the vision encoder before and after VLM pretraining and reveal that the decline in linear decodability is the primary driver of the observed degradation in social representations. To solve this problem, the author proposed SocialFusion with native support for bounding box and heatmap generation for fusing diverse visual social interaction understanding tasks. The result achieves positive transfer on all the tasks while the other baselines suffer from negative transfer for most of the tasks.

**Additional Comments:**

In summary, I think the authors demonstrate that VLM pretraining harms the capabilities of social interaction understanding for some lightweight VLMs. I think more experiments could be incorporated to demonstrate that this is a common problem in the state-of-the-art VLMs. Besides, I have concerns about the proposed solutions. Can it keep its original capabilities on other tasks? Or it is only designed for social interaction and understanding tasks. I hope to see more results in the author's revised version.

**Audience:**

Yes

**Audience Explanation:**

Current state-of-the-art VLMs usually have limited discussions on their capabilities for social interaction understanding tasks. Although there are some works pointing out that the VLMs have limitations in social interaction understandings, currently, no work has studied why it could happen, as far as I know. Although this submission only provides one possible cause for this problem (other reasons, such as the number of parameters and data), it still brings some insight to the community, and leaves some of the future research topics. Therefore, I think some individuals in TMLR’s audience will be interested in this work.

**Broader Impact Concerns:**

I don’t see aspects of the submission that could cause broader impact concerns.

**Claims And Evidence:**

Yes

**Claims Explanation:**

The authors claim that the social degradation of pretrained VLMs is caused by the vision-linguistic alignment in the pretrained stage for VLMs. To support this claim, the authors conduct comprehensive controlled experiments. They test the decodability of the vision encoder before and after the VLM pretraining by linear representation probes. The result demonstrates that the vision encoder after VLM pretraining is less decodable than the vision encoder before VLM pretraining. This suggests that the VLM has harm the vision encoder’s capabilities for social interaction and understanding tasks. The author also conducts experiments on compatibility among different tasks of social interaction and understanding. The result demonstrates that the diversity of the gradient differs a lot among the tasks. However, the difference is marginal. Based on the content of their experiment sections, I think the claims are well supported.

**Requested Changes:**

I appreciate that the authors have demonstrated the potential problem of VLM pretraining in social interaction understanding tasks, I have the following concerns that may need to be addressed in your revised version. The significant of my concerns are from most to the least.
1.	Only 3 baselines are limited to support your claims. Besides, Qwen2 and SAIL-VL-1.5B were posted last year, which are out-of-date as the fast development of VLMs in recent years. Qwen2.5-VL has already been released for several months and InternVL also has 1B version. These VLMs are more powerful than the baselines that authors include in the submissions. Therefore, we don’t know whether this problem has already been solved by those state-of-the-art VLMs. I would like to see whether we can have similar observations in these methods.
2.	I understand that the authors use VLMs with a parameter size of 2B for fair comparison with your SocialFusion Framework. However, 2B VLMs are usually designed for on-device purposes, which usually have limited understanding capabilities and are not used for complicate task such as social interaction understanding. The scale of data they use to train those models is usually not comparable to the VLMs with more parameters. As you demonstrated in your experiments, decodability drops after VLM pretraining. Another possible hypothesis is that limited and noisy social interactions data in the pretraining stage weaken the prior knowledge of the vision encoder. I’m curious whether we can observe similar problems for some larger state-of-the-art models, such as Qwen3-VL-8B, which are designed for the usage of most people.
3.	The proposed solution of SocialFusion seems to solve the problem of the model capabilities in social interaction understanding. However, the paradigm of frozen vision encoder with a learnable adapter is the approach of early-stage VLMs, such as the original version of LLaVA (Neurips 2023). Researchers finetune vision encoder in follow-up work because it can enhance its ability to capture local details and contextual relationships and achieve better performance on complex tasks such as OCR. Will SocialFusion harm the performance of those downstream tasks?
4.	Some minor issues with the readability of tables. The Column with “Vision Encoder” is actually the model’s name which brings confusion when I read it.

---

> ### Author Response · Authors · 2025-11-21
> **Response**
>
> We thank the reviewer for the constructive review and address your concerns below.
>
> ---
>
> > Only 3 baselines are limited to support your claims. [...] I would like to see whether we can have similar observations in these methods.
>
> Though our baseline VLMs are no longer the newest state of the art, we remain confident in our findings involving these models. The models we chose represent contrasting architectural and data decisions, with MolmoE being a mixture-of-experts model (7b total parameters), Sail-VL being focused on high quality data curation, and Qwen2-VL focusing on compressing the visual information as much as possible. By demonstrating the same findings on all three of these contrasting approaches, we show that the problem we identify is general.
>
> However, your point that the fundamental weakness may no longer be present in newer models is still important. To address this, we present joint vs single task training results for InternVL 3, which was released earlier this year and represents yet another set of design choices. We added the following table to the paper:
>
> | Model | Joint Training | HaGRIDv2 | HaGRIDv2 | PISC | PISC | LAM | LAM | GazeFollow | GazeFollow | AffectNet | AffectNet |
> |---|---|---|---|---|---|---|---|---|---|---|---|
> |  |  | mAP | Acc | Domain | Relation | mAP | Acc | Min L2 | Max L2 | mAP | Acc |
> | InternVL3 | ✗ | **99.8** | **98.5** | **95.5** | 91.9 | **82.3** | **93.8** | **0.070** | **0.132** | **65.9** | **51.8** |
> | InternVL3 | ✓ | 99.5 | 97.3 | **95.5** | **92.1** | 81.1 | 93.7 | 0.081 | 0.145 | 63.2 | 47.9 |
>
> We observe negative transfer on 8/10 metrics, showing that newer VLMs at this scale still struggle with jointly modeling visual social understanding tasks. While computation limitations prevent us from performing all experiments on this model, these results show that this critical weakness (which SocialFusion does not exhibit) remains a relevant and timely issue.
>
> > I understand that the authors use VLMs with a parameter size of 2B for fair comparison with your SocialFusion Framework. However, 2B VLMs are usually designed for on-device purposes, which usually have limited understanding capabilities and are not used for complicate task such as social interaction understanding.
>
> Beyond allowing for faster experimentation, we argue that this model scale is extremely relevant for studying social interaction understanding. First, as you note, these models are primarily intended to be deployed on-device, where they must be useful in everyday life. When deployed to augmented reality devices like smart glasses, these models must understand the social situation the user is in to provide the best possible assistance. Users will be expecting it to understand basic social dynamics in order to answer relevant questions. Similarly, on a phone, many of a user’s questions may involve basic social understanding, such as interpreting the subtext behind a message.
>
> You hypothesize that at this scale, “limited and noisy social interactions data in the pretraining stage weaken the prior knowledge of the vision encoder.” If this is the case, it is critical for future on-device models to address this weakness in data curation to improve social understanding.
>
> > The proposed solution of SocialFusion seems to solve the problem of the model capabilities in social interaction understanding. [...] Will SocialFusion harm the performance of those downstream tasks?
>
> SocialFusion is not applicable to arbitrary tasks because its visual-linguistic connector is trained from scratch on only social tasks. While this is a limitation compared to off-the-shelf VLMs, SocialFusion is not intended for unseen tasks. Instead, its purpose is to 1) demonstrate that the VLM paradigm is not inherently incapable of unifying and exhibiting positive transfer on diverse visual social interaction understanding tasks and 2) study the synergies between different social tasks and the degradation of VLM encoders. As such, it would not be productive to measure its performance on other tasks, but future research should investigate general VLMs that do not suffer from this social degradation. We have made this point clearer in the revised manuscript.
>
> > Some minor issues with the readability of tables. The Column with “Vision Encoder” is actually the model’s name which brings confusion when I read it.
>
> This column indicates the *source* of the visual encoder. It must be read in conjunction with the “VLM Pre-training” column to identify the specific model. For example, “MolmoE” + “VLM Pre-training” is the MolmoE visual encoder, but “MolmoE” without “VLM Pretraining” is CLIP, which is the encoder that MolmoE fine-tuned.
>
> ---
>
> Thank you again for your review. We hope we have addressed your concerns and are happy to provide further clarification.

---

### Decision · Action_Editor_nqtc · 2025-12-24

**Recommendation:** Accept as is

**Additional Comments:**

After the authors' revision, most of the reviewers' concerns have been addressed. The reviewers unanimously recommend acceptance of the paper.

I find the work valuable and recommend its acceptance and publication.

As they prepare the final version, I would suggest that the authors move and further enrich the experimental results on InternVL3 from the appendix to the main paper. If possible, including evaluations on more recent and advanced VLMs would also help demonstrate that the issues investigated in this work persist even for state-of-the-art models, thereby highlighting the broader implications of the study.

**Audience:**

Yes

**Audience Explanation:**

This paper is likely to be of interest to researchers and practitioners working on social interaction understanding, transfer learning, and the application of pre-trained foundation models to downstream tasks.

**Claims And Evidence:**

Yes

**Claims Explanation:**

Summary:

This paper analyzes why modern vision-language models (VLMs) often fail to achieve positive transfer in visual social interaction understanding despite the strong similarity among tasks. Through controlled multi-task experiments on five social perception tasks, a “social degradation” phenomenon is identified where standard VLM pre-training harms the visual encoder’s ability to represent social cues. Using linear probing and gradient analysis, reduced feature decodability after VLM pre-training is shown as the primary cause of this degradation. To address this issue, the paper proposes SocialFusion, a unified architecture that freezes the visual encoder and language model and learns a lightweight visual-language adapter supporting both classification and heatmap-based outputs. Experiments demonstrate consistent positive transfer across all tasks and metrics, outperforming standard VLM fine-tuning and matching task-specific state of the art on several benchmarks. Overall, the work highlights a fundamental limitation of current VLM pre-training for social understanding and presents a practical design that restores cross-task synergy.


Claims:

The paper makes several key claims: (1) standard vision-language model (VLM) pre-training induces social degradation, where visual encoders lose their ability to represent fine-grained social cues, leading to negative transfer across social interaction understanding tasks; (2) this degradation is driven primarily by reduced linear decodability of social representations after VLM pre-training, with gradient conflicts across tasks playing a secondary role; (3) freezing the visual encoder and learning only a minimal visual-language connector is an effective strategy for unifying diverse social perception tasks; and (4) the proposed SocialFusion architecture consistently achieves positive transfer across all evaluated social tasks and establishes new state-of-the-art results on HaGRIDv2 and PISC, demonstrating a practical path toward generalist visual social interaction understanding.


Evidence:

The empirical evidence presented in the paper largely supports the stated claims, although evaluations with more recent and advanced VLMs, as well as comparisons against more sophisticated baselines, remain limited. These limitations are appropriately acknowledged and addressed in the revised version.